# Antiviral Potentialities of Chemical Characterized Essential Oils of *Acacia nilotica* Bark and Fruits against Hepatitis A and Herpes Simplex Viruses: *In Vitro*, *In Silico,* and Molecular Dynamics Studies

**DOI:** 10.3390/plants11212889

**Published:** 2022-10-28

**Authors:** Abd El-Nasser G. El Gendy, Ahmed F. Essa, Ahmed A. El-Rashedy, Abdelbaset M. Elgamal, Doaa D. Khalaf, Emad M. Hassan, Ahmed M. Abd-ElGawad, Abdallah M. Elgorban, Nouf S. Zaghloul, Salman F. Alamery, Abdelsamed I. Elshamy

**Affiliations:** 1Medicinal and Aromatic Plants Research Department, National Research Centre, Dokki, Giza 12622, Egypt; 2Department of Natural Compounds Chemistry, National Research Centre, Dokki, Giza 12622, Egypt; 3Natural and Microbial Products Department, National Research Centre, Dokki, Giza 12622, Egypt; 4Department of Chemistry of Microbial and Natural Products, National Research Centre, Dokki, Giza 12622, Egypt; 5Department of Microbiology and Immunology, National Research Centre, Dokki, Giza 12622, Egypt; 6Department of Botany, Faculty of Science, Mansoura University, Mansoura 35516, Egypt; 7Department of Botany and Microbiology, College of Science, King Saud University, P.O. Box 2455, Riyadh 11451, Saudi Arabia; 8Bristol Centre for Functional Nanomaterials, HH Wills Physics Laboratory, Tyndall Avenue, Bristol BS8 1FD, UK; 9Biochemistry Department, College of Science, King Saud University, P.O. Box 2455, Riyadh 11451, Saudi Arabia

**Keywords:** gum arabic tree, volatile compounds, antiviral activity, in silico, stachene

## Abstract

*Acacia nilotica* (synonym: *Vachellia nilotica* (L.) P.J.H.Hurter and Mabb.) is considered an important plant of the family Fabaceae that is used in traditional medicine in many countries all over the world. In this work, the antiviral potentialities of the chemically characterized essential oils (EOs) obtained from the bark and fruits of *A. nilotica* were assessed in vitro against HAV, HSV1, and HSV2. Additionally, the in silico evaluation of the main compounds in both EOs was carried out against the two proteins, 3C protease of HAV and thymidine kinase (TK) of HSV. The chemical profiling of the bark EOs revealed the identification of 32 compounds with an abundance of di- (54.60%) and sesquiterpenes (39.81%). Stachene (48.34%), caryophyllene oxide (19.11%), and spathulenol (4.74%) represented the main identified constituents of bark EO. However, 26 components from fruit EO were assigned, with the majority of mono- (63.32%) and sesquiterpenes (34.91%), where *trans*-caryophyllene (36.95%), *Z*-anethole (22.87%), and γ-terpinene (7.35%) represented the majors. The maximum non-toxic concentration (MNTC) of the bark and fruits EOs was found at 500 and 1000 µg/mL, respectively. Using the MTT assay, the bark EO exhibited moderate antiviral activity with effects of 47.26% and 35.98% and a selectivity index (SI) of 2.3 and 1.6 against HAV and HSV1, respectively. However, weak activity was observed via the fruits EO with respective SI values of 3.8, 5.7, and 1.6 against HAV, HSV1, and HSV2. The in silico results exhibited that caryophyllene oxide and spathulenol (the main bark EO constituents) showed the best affinities (ΔG = −5.62, −5.33, −6.90, and −6.76 kcal/mol) for 3C protease and TK, respectively. While caryophyllene (the major fruit EO component) revealed promising binding capabilities against both proteins (ΔG = −5.31, −6.58 kcal/mol, respectively). The molecular dynamics simulation results revealed that caryophyllene oxide has the most positive van der Waals energy interaction with 3C protease and TK with significant binding free energies. Although these findings supported the antiviral potentialities of the EOs, especially bark EO, the in vivo assessment should be tested in the intraoral examination for these EOs and/or their main constituents.

## 1. Introduction

The herpes simplex virus (HSV, family: *Herpesviridae*) is a pathogenic virus that causes a wide range of human infections around the world [1,2]. The HSV types, including (HSV-1 and -2) cause several diseases such as oral cavity infections, eyes, pharynx, and mucous membrane, especially in immunodeficient patients [3,4]. The current usual anti-HSV drugs are described to have several side effects and a high prevalence of resistance [5]. This trouble strongly stimulated scientists to seek new therapeutic drugs and agents to overcome virus resistance with low or no side effects. Natural products, including different extracts, essential oils (EOs), and metabolites of herbal and medicinal plants, represented the main strategy for dissolving these issues [6,7].

The hepatitis A virus (HAV) is one of the causes of more than a million infections around the world every year. This virus is a liver-self-limiting and sometimes life-threatening disease [8]. HAV infections were primarily caused by foods and water, as well as direct contact with an infected person [8]. HAV is the most endemic disease in several developing countries [9]. 

Several plant extracts and metabolites have been documented to have the ability to protect and/or treat these types of viruses. Tannins such as chebulinic acid and chebulagic acid have been shown to have anti-HSV-2 activity nearly 20 times greater than acyclovir, with greater selectivity [10]. In addition, flavonoid aglycones including chrysin, fisetin, and galangin as well as prenylated flavonoids such as kuwanon T exhibited 4–3 more potent activity against HSV-1 than acyclovir [11,12,13]. Several EOs derived from ginger, thyme, hyssop, anise, sandalwood, and camomile demonstrated significant potencies against herpes simplex virus [14,15]. Additionally, geraniol, a metabolite derived from essential oils, showed a promising anti-HSV-2 activity similar to that of acyclovir [16]. 

The 3C protease enzyme is one of the main participants in the process of viral genome replication. This enzyme cleaves the viral polyprotein into capsid and nonstructural proteins and inhibits IRES-dependent translation, enabling subsequent viral replication [17]. This motivated researchers to target this protein to inhibit the viral activity. In addition, thymidine kinase, one of the virulence-related proteins in viruses, plays a vital role in the steps of the phosphorylation process associated with the nucleoside salvage pathway. Moreover, this protein can convert pro-antiviral nucleoside analogs such as acyclovir and its congeners to their active forms [18].

The EOs of plants are complicated mixtures of compounds of low molecular weights that are extracted from different medicinal and aromatic plants via different tools [19]. They included many metabolites belonging to different chemical classes, including terpenes and phenylpropanoids, representing the main components along with traces of other aliphatic and aromatic constituents [19,20]. Many biological properties of EOs have been discovered, including antimicrobial, cytotoxicity, anti-inflammatory, antipyretic, antiulcer, and antiviral properties [19].

*Acacia* plants (family: Leguminosae or Fabaceae) include around 1350 species or more that are widely distributed in warm and tropical areas. Several *Acacia* species have been shown to have pharmaceutical effects, including the treatment and inhibition of microbial infections, ulcerative colitis, skin and eye infections, wounds, mouthwash, and others [21,22,23]. Many compounds were documented from the plants belonging to the *Acacia* genus, especially di- and triterpenes, flavonoids, tannins, alkaloids, phenolic acids, and saponins [21,23]. The EOs were also reported from different *Acacia* plants, such as the Nigerian *Acacia nilotica* (L.) *P.J.H.Hurter and Mabb.*, *A. albida* (Delile) A.Chev. (synonym: *Faidherbia albida*) [24], *A. tortilis* (synonym: *Vachellia tortilis* (Forssk.) Galasso and Banfi) [25], *A. mearnsii* De Wild. (synonym: *A. mearnsii*) [26], and *A. cyanophylla* (synonym: *Acacia saligna* (Labill.) H.L.Wendl.) [27].

*Acacia nilotica* (common name: Gum Arabic tree) is an important traditional plant for the treatment of several diseases in several countries around the world [22]. Many components were isolated and identified from the extracts of the different organs of this plant comprising the terpenes, tannins, phenolic acids, and flavonoids [21,22,28,29]. Several reports described the interesting biological potentialities of the different extracts of this plant, such as the treatment of inflammation, free radicals, leishmanial, diabetes, cancers, plasmodial, and other infections, along with molluscicidal activity [30,31,32].

Little studies were carried out concerning the composition and pharmaceutical applications of *A. nilotica* EOs [24]. Thus, the current work aimed to (i) characterize the chemical profile of the EOs derived from the *A. nilotica* bark and fruits depending upon GC-MS techniques, (ii) evaluate the antiviral efficiency of these two EOs against the HSV1, HSV2, and HAV viruses, and (iii) study of the molecular docking of the main compounds in each oil on 3C protease of HAV (PDB ID: 1QA7) and thymidine kinase of HSV (PDB ID: 1KI3) proteins.

## 2. Results and Discussion

### 2.1. The Identification of the Chemical Constituents of Bark and Fruits of A. nilotica EOs

The EOs of the bark and fruits of *A. nilotica* were separately obtained via three hours of hydro-distillation over the Clevenger apparatus, where they yielded 0.072% and 0.056% (*v*/*w*) of the oil, respectively. The yield of the EO from bark was found to be more than the fruits, and both were more varied than those described for the Nigeria ecospecies [24]. The oils were analyzed via GC-MS (Figure 1).

The compounds’ names, retention times, relative concentrations, and Kovats indexes are presented in Table 1. The analysis revealed that the terpenes represented the main constituents of the bark with a relative concentration of 95.25% of the hydrocarbons and oxygenated forms of mono-, sesquit-, and di-terpenes. Furthermore, the terpenes represented the major components of the fruits EO with a relative concentration of 98.23%, including the mono- and sesquiterpene hydrocarbons along with the oxygenated derivatives of monoterpenes. This preponderance of terpenes agreed with the published data from the Nigerian *A. nilotica* [24] and other *Acacia* species such as *A. albida* [24], *A. tortilis* [25], *A. mearnsii* [26], and *A. cyanophylla* [27].

The chemical characterization of the bark EO revealed that the diterpenes are the major constituents with a relative concentration of 54.60%, comprising of diterpene hydrocarbons (53.54%) and traces of oxygenated diterpene (1.06%). With a few exceptions, such as *Euphorbia mauritanica* L. [33], *Lactuca serriola* L. [34], and others [35], the phenomenon of diterpene abundance was rare in plants EOs. Four diterpene hydrocarbons were assigned as overall identified diterpenoids, including stachene (48.34%), trachyloban (2.25%), kaur-16-ene (1.94%), and isokaurene (1.01%). Stachene was reported as a major diterpene hydrocarbon in a number of plants, including *Chamaecyparis pisifera* (Siebold and Zucc.) Endl., *Chamaecyparis obtuse* (Siebold and Zucc.) Endl., and *Thuja orientalis* L. [36].

Trachylobane and kaur-16-ene were rarely found in *Acacia* EOs but were widely found in the EOs of other plants, such as *Croton zambesicus* Muell. Arg. [37], *Cryptomeria japonica* (L.f.) D.Don [38], *Alpinia galanga* (L.) Willd. [39], *Euphorbia heterophylla* L. [40], and *E. mauritanica* [33]. Three oxygenated diterpenes, phytol, sclareol, and 4,8,13-duvatriene-1,3-diol, were identified from the bark. The phytol was documented as the main component of EO derived from the leaves of *Acacia mearnsii* [26]. These three compounds are well-known and commonly reported compounds in the EOs of some plants, such as *Calotropis procera* (Aiton) W.T.Aiton [41], *Cyperus leavigatus* L. [42], and *Launaea mucronata* (Forssk.) Muschl. [43]. On the contrary, the GC-MS analysis of the EO derived from the fruits revealed the complete absence of the diterpenoid constituents. The phenomena of the diterpenes’ disappearance from the EOs of *Acacia* species is commonly documented [19,24,25,27]; these data agree with the data of fruit oil but not with the bark oil.

Monoterpenes were found to be the major characterized components of *A. nilotica* fruits EO, with a relative concentration of 63.32% of mainly oxygenated compounds (55.51%) and hydrocarbons (7.81%). The abundance of the monoterpenoids in the fruits EO agreed with the previous data from Nigerian *A. nilotica* and other *Acacia* species [24,26,27,33]. Z-Anethole (22.87%), 2-caren-10-al (3.51%), and borneol (2.31%) were characterized as the main identified oxygenated monoterpenes, while γ-terpinene (7.35%) was the major monoterpene hydrocarbons while 1,8-cineol (3.52%) was assigned. All these compounds were reported in the EOs of *Acacia* species as traces and/or totally absent [19,24,26,33]. However, the bark EO was found to have traces of monoterpenes in oxygenated forms.

The bark and fruits EOs were found to include the sesquiterpenes with high relative concentrations (39.81% and 34.91%, respectively). The analysis of the bark EO revealed the presence of a low concentration of sesquiterpene hydrocarbons (8.27%) and major oxygenated sesquiterpenes (31.54%). The α-muurolene (2.42%) and bicyclogermacrene (2.41%) were identified as the major sesquiterpene hydrocarbons, while caryophyllene oxide (19.11%) and spathulenol (4.74%) are the main oxygenated sesquiterpenes of the bark EO. Furthermore, the sesquiterpene hydrocarbons were the only detected sesquiterpenes in the EO of the fruits with a relative concentration of 34.91% and a complete absence of oxygenated compounds. From all the assigned sesquiterpene hydrocarbons in fruits EO, *trans*-caryophyllene (36.95%), α-elemene (4.69%), α-humulene (4.05%), and β-elemene (3.72%) represented the main constituents. The majority of the sesquiterpenes were previously reported from EOs of the Nigerian *A. nilotica* [24], and other *Acacia* plants as *A. tortilis* [25]. Previous studies of EOs derived from *Acacia* species revealed the prevalence of muurolene, caryophyllene oxide, and spathulenol, where they are determined in considerable concentrations [24,25,26].

Ultimately, the other non-terpenoid compounds are represented as traces of the total mass of the bark EO (3.49%), while they are absent in fruit EO. In bark EO, *n*-nonacosane (2.51%) and *n*-dotriacontane (0.98%) were identified. Hydrocarbons are widely identified compounds in *Acacia* plants [25,26]. As observed in these findings, the net results revealed a significant variation in the quantity and quality as well as the chemical components of the two plant parts and also in Nigerian *A. nilotica* [24]. This phenomenon of variation might be attributed to the variation of the plant organ, genotypes, humidity, climate, weather, and environmental conditions [41,44].

### 2.2. In Vitro Antiviral Activity

The antiviral activities of EOs derived from the bark and fruits of *A. nilotica* were in vitro screened against HAV, HSV1, and HSV2, and the maximum non-toxic concentration (MNTC) used for the screening was determined. The results showed that MNTC of EOs of bark and fruits was 500 and 1000 µg/mL, respectively (Table 2). Using the MTT assay, the percentages of antiviral effects of bark and fruits EOs were determined by comparing the viability of cells treated by the virus only with the viability of cells treated by the virus and MNTC of the samples.

The results revealed that bark EO has moderate antiviral effects against HAV with an effect of 47.26% alongside a selectivity index (SI) of 2.3, concerning amantadine as a reference drug with SI at 51.62. In addition, the bark EO demonstrated moderate antiviral abilities against HSV1 with an effect of 35.98% and SI of 1.6 compared with acyclovir as a positive control at SI > 387.63. However, this EO exhibited a very weak antiviral effect against HSV2 with a 9.07% comparison with acyclovir with SI at 12.24.

On the other side, the EO derived from the fruits showed weak potentialities against the three tested viruses. This oil showed weak anti-HAV with an effect % of 9.42% and SI of 3.8 compared with amantadine at SI of 51.62. Likewise, this oil also exhibited weak anti-HSV1 with effect% at 14.26% and SI of 5.7 compared with acyclovir as a standard drug. In addition, the results revealed that fruits EO has the lowest activity against HSV2 with an effect% of 3.99% as well as SI at 1.6.

The chemical components of the EOs and especially the main compounds played as the main mediators in their pharmaceutical and biological activities [8]. The present results revealed that the bark EO has moderate to weak antiviral potentialities, especially against the two viruses, HAV and HSV1. The GC-MS profiling of this oil revealed the abundance of terpenes, mainly di- and sesqui-terpenes, that were documented to have antiviral effects against several viruses [45,46]. Diterpenes, as the main components of the bark EO, were reported to display significant antiviral effects against HAV and HSV [45,47,48,49].

Plants’ EOs enriched with diterpenes were stated to demonstrate significant antiviral agents such as the different extracts and EOs of *Croton lechleri* Müll.Arg. [2]. Mechanistically, the diterpenes could inhibit the viral replication process by blocking DNA polymerase activity, as reported with dolastane compounds [49]. Several diterpene skeletons, including kaurenes, have been shown to have significant anti-HSV, anti-HAV, and other antiviral properties [45,49]. In addition, the sesquiterpenes acted in a significant role as inhibitors of several viruses. Astani and his colleagues discovered that caryophyllene oxide has significant anti-HSV1 activity [47]. Many EOs derived from plants, such as *Eryngium alpinum* L., *Eryngium amethystinum* L. [50], *Melaleuca ericifolia* Sm., *Melaleuc leucadendron* (L.) L., *Melaleuc armillaris* (Sol. ex Gaertn.) Sm. and *Melaleuc styphelioides* Sm. [51], have been shown to have antiviral activities due to the majority of caryophyllene oxide and spathulenol. The two sesquiterpenoids, α-muurolene and bicyclogermacrene, were also reported as major components of antiviral active EOs derived from *Glechon spathulata* Benth. and *Glechon marifolia* Benth. [52].

Previous studies hypothesized that the anti-HSV effectiveness greatly depends on the binding affinity of EO components to the surface of viruses, and this affinity might be affected by the polarity of compounds [15]. The anti-HSV effects of thymol-related monoterpenoids were reported to decrease with decreasing polarity [53].

Consequently, the present findings were in full agreement with the previous reports. The singular and/or synergetic effects of the major components, along with the minors of the bark EO, might be the main reasons for this anti-HAV and HSV1 efficiency. These results supported the ability of these compounds to cause abnormalities in the HSV membrane protein functions and structures that consequently decrease the penetration or binding of the virus into the cells [54].

On the other hand, the fruits’ EO antiviral effects were in concordance with Astani et al. [47] who deduced the anti-HSV with β-caryophyllene and Z-anethole as singular compounds. The increase in the HSV inhibitory effects of the EOs was basically ascribed to the presence of the polyhydroxylated constituents [54]. With the exception of the mono-oxygenated monoterpene, Z-*anethole*, the other oxygenated monoterpenes were identified as traces that decrease the contribution and effect of each singular compound. Several plants, including *Mentha aquatic* L., *Mentha pulegium* L., *Mentha microphylla* C. Kock, Mentha x villosa (Hudson), *Micromeria thymifolia* (Scop.) Fritsch, and *Ziziphora clinopodioides* Lam., have demonstrated weak antiviral activity of EOs enriched with oxygenated monoterpenes [55].

### 2.3. Molecular Docking Studies

In our attempt to study the mechanistic action of each EO, the molecular operating environment (MOE) of the main three compounds in each EO toward the two proteins, HAV 3C protease and HSV TK, were evaluated. The results of in silico studies of the major compounds identified in both oils could rationalize the in vitro antiviral activities noticed by them. As shown in Table 3, caryophyllene oxide and spathulenol, the main constituents of oil of the bark, showed the best affinities (ΔG = −5.62, −5.33 and −6.90, −6.76 kcal/mol) for 3C protease and TK, respectively (Figure 2 and Figure 3). Additionally, trans-caryophyllene, as a major constituent of fruit EO, revealed promising binding capabilities against both proteins (ΔG = −5.31, −6.58 kcal/mol, respectively) (Figure 2 and Figure 3).

Previous studies have shown that the sesquiterpene caryophyllene and its oxides possess suitable antiviral activities with a high selectivity index [47]. The higher activity of the bark oil than the fruit oil might be attributed to the synergistic effect of caryophyllene oxide and spathulenol, as well as the other components. These data go in the same line with the in vitro results in which the bark EO is more active than the fruits EO.

### 2.4. Molecular Dynamic and System Stability

A molecular dynamic simulation was carried out to predict the inhibition performance and interaction of the caryophyllene oxide with the catalytic active site of both 3C protease of HAV and thymidine kinase of HSV targets as well as their stability throughout the simulation [56,57]. The validation of system stability is essential to trace disrupted motions and avoid artifacts that may develop during the simulation. The recorded average RMSD values for all frames of Apo-HAV and Complex-HAV were 1.53 Å and 1.37 Å, respectively. In addition, average RMSD values of 1.51 Å and 1.59 Å were observed for Apo-HSV and Complex-HSV, respectively (Figure 4A,C). In general, These results revealed that the caryophyllene oxide-bound protein complex system acquired a relatively more stable conformation than the other studied systems.

During MD simulation, assessing protein structural flexibility upon ligand binding is critical for examining residue behavior and its connection with the ligand. The protein residue fluctuations were evaluated using the Root-Mean-Square Fluctuation (RMSF) algorithm to evaluate the effect of inhibitor binding toward the respective targets over 20 ns simulations. The computed average RMSF values were 1.21 and 1.00 for Apo-HAV, and caryophyllene oxide-bound to protein systems, respectively, while 1.17 Å and 1.02 Å were recorded for Apo-HSV and Complex-HSV, respectively. Figure 4B,D show the overall residue fluctuations of individual systems. These values reveal that the caryophyllene oxide-bound protein complex system has a lower residue fluctuation than the other systems.

#### Binding Interaction Mechanism Based on Binding Free Energy Calculation

A popular method for determining the free binding energies of small molecules to biological macromolecules is the molecular mechanics energy technique (MM/GBSA), which combines the generalized born and surface area continuum solvation, and it may be more trustworthy than docking scores. The MM-GBSA program in AMBER18 was used to calculate the binding free energies by extracting snapshots from the trajectories of the systems. As shown in Table 1, all the reported calculated energy components (except ΔG_solv_) gave negative values, indicating favorable interactions. The results indicated that binding free energy (ΔG_bind_) values −19.35 and −32.04 Kcal/mol were obtained for the interactions of caryophyllene oxide with 3C protease of HAV and thymidine kinase of HSV receptors, respectively.

A close look at the individual contributions of energy reveals that the more positive van der Waals energy components drive caryophyllene oxide interactions with 3C protease of HAV and thymidine kinase of HSV enzyme, resulting in the observed binding free energies (Table 4).

## 3. Materials and Methods

### 3.1. Plant Materials and EOs Extraction

*Acacia nilotica* bark and fruits were collected in August 2020 from Naje Algamal, Sohag Governorate, Egypt (26°33′43.2″ N 31°40′39.7″ E). The specimen is authenticated and deposited in the herbarium of the National Research Center with specimen codes NRC-20xYD-AN-20-609.

The collected samples were dried at air temperature (25 °C ± 3) for 7 days and then crushed into a fine powder. The EOs of the air-dried powdered bark and fruits of *A. nilotica* (300 g, each) were subjected to hydro-distillation over a Clevenger apparatus for three hours. The oily layers were separated via *n*-hexane and then dried using Na_2_SO_4_ anhydrous (0.5 g). Three EO samples from each part were obtained by repeating the extraction process three times with the same sequence. All oil samples were saved in glass vials at 4 °C until further GC-MS and biological analysis.

### 3.2. Gas Chromatography-Mass Spectrometry (GC-MS) Analysis

The GC-MS technique was used for the analysis of the extracted EO samples, as reported in published studies [33,43]. The analysis was performed at the National Research Center using the GC-MS apparatus joined with the TRACE GC Ultra-Gas Chromatograph and quadrupole mass unit, model Thermo-Scientific™ EC, Waltham, MA, USA. The used GC-MS column dimension and film thicknesses were 0.32 mm × 30 m and 0.25 µm. The transporter gas (He) was used with 1 to 10 as the split ratio and 1.0 mL/min as the flow rate. As usual, the temperature was regulated as follows: 60 °C/1 min and elevated to 240 °C during 4 °C/min. The dilution of all oil samples was performed in 1 µL of n-hexane by the ratio of 1:10 (*v*/*v*) and injected where both of injector and detector were adjusted at 210 °C. The electron ionization (EI) at 70 eV and *m*/*z* 40–450 as spectral range were used for performing the mass spectral data of oil constituents. Finally, the authentication and identification of the chemical constituents occurred depending upon the Automated Mass spectral Deconvolution and Identification (AMDIS) software, (version: 1.0.0.13), NIST library database, Wiley spectral library collection, and retention indices relative to *n*-alkanes (C_8_–C_22_).

### 3.3. Antiviral Assays

#### 3.3.1. Determination of the Maximum Non-Toxic Concentration (MNTC)

Solutions of 10 mg of EOs of bark and fruits of *A. nilotica* were prepared in 10% dimethylsulfoxide (DMSO) and then diluted with Dulbecco’s Modified Eagle Medium (DMEM) (Lonza, Verviers, Belgium) to the working solutions. For the modified 3-(4, 5-dimethylthiazol-2-yl)-2,5-diphenyltetrazolium bromide (MTT) assay, the VERO-E6 cells (3 × 10^5^ cells/mL) (Sigma-Aldrich, St. Louis, MO, USA) were seeded in 96-well plates (100 µL/well) and incubated in 5% CO_2_ at 37 °C. After 24 h of incubation, cells were subjected to concentrations (31.25, 62.5, 125, 250, 500, and 1000 µg/mL) of EOs of bark and fruits of *A. nilotica* in triplicates for another 48 h. Cells were checked for any physical signs of toxicity, e.g., partial or complete loss of the monolayer, rounding, shrinkage, or cell granulation. Then, the cell monolayers were washed 3 times with 1 L sterile phosphate buffer saline (PBS) and treated with 20 µL MTT solution in PBS (5 mg/mL) for each well. After shaking, the plate was incubated for 4 h in 5% CO_2_ at 37 °C. The formed formazan crystals in each well were dissolved with 200 µL of 0.04 M HCl in isopropanol. Optical density was determined at 560 nm after removing the background at 620 nm, where it should be directly correlated with cell quantity. The maximum non-toxic concentration (MNTC) of each extract was determined and was used for further biological studies. The MNTC was calculated from the plot of toxicity percent against sample concentration. The DMSO was tested as a control, and it did not show any activity.

#### 3.3.2. Antiviral Effect Percent Determination

All viruses, HSV1, HSV2, and HAV, were obtained from the Microbiology Department, Faculty of Medicine for girls, Al-Azhar University. Vero-E6 cells were distributed in 96-well tissue culture plates (10^4^/well) and incubated at a humidified 37 °C incubator under 5% CO_2_ conditions. After 24 h, the cells were washed with 1 L of PBS. Equal volumes (1:1 *v*/*v*) of MNTC of tested samples and the virus (HSV1, HSV2, and HAV) suspension were incubated for one hour. In triplicate, 100 L of viral or sample suspension was propagated on cells and incubated for 24 h at 37 °C in 5% CO_2_. A total of 20 µL MTT solutions in PBS (5 mg/mL) was added to each well and incubated for 1–5 h to allow MTT to be metabolized. As previously mentioned, the resultant was treated, and the optical density was determined. The antiviral effect percentage was calculated by dividing the viability of cells treated by the virus only/the viability of cells treated by the virus and the sample. Acyclovir was used as a reference drug for the viruses (HSV1 and HSV2), while amantadine was used as a positive control against the HAV virus. The calculation of the SI (selectivity index) values of the tested EO samples was performed from the equation: SI = CC_50_/IC_50_; where: CC_50_: 50% cytotoxic concentration, and IC_50_: 50% effective concentration [6,11].

### 3.4. Molecular Docking Studies

Molecular Operating Environment (MOE Vs. 2015) docking studies were performed on the catalytic domains of HAV 3C protease (PDB ID: 1QA7) [58] and HSV thymidine kinase (PDB ID: 1KI3) [58] and thymidine kinase of HSV (PDB ID: 1KI3) [59]. The crystal structures of both proteins were retrieved from the protein data bank and processed as previously described [60], as well as the database file (mdb) of the major identified compounds in both EOs, stachene, caryophyllene oxide, spathulenol, *trans*-caryophyllene, *Z*-anethole, and *γ*-terpinene. The docking process was validated by re-docking the co-crystallized ligands in the binding site, which revealed their binding with crucial sub-pockets (Val 28 and Cys 172 and Gln A125, Arg A176, and Glu A83) in both proteins, respectively, at acceptable RMSD (Figure 2 and Figure 3). After processing the study in triplicate (as shown in Appendix A), the results of the docking process were presented as the ΔG (kcal/mol) with RMSD values ≤ 2 Å (Table 2). In addition, the interactions of the lowest energy pose with the binding pocket were two and three-dimensionally presented in (Figure 2 and Figure 3).

### 3.5. Molecular Dynamics Stimulation Section

#### 3.5.1. System Preparation

The crystal structures of the 3c proteinase from the hepatitis A virus receptor and the thymidine kinase from herpes simplex virus type I were retrieved from the protein data bank with codes 1QA7 [58] and 1KI3 [59], respectively. This structure was then prepared for molecular dynamics (MD) studies using UCSF Chimera [61]. Using PROPKA, pH was fixed and optimized to 7.5 (3). Caryophyllene oxide was drawn using ChemBioDraw Ultra 12.1. Altogether, all four prepared systems were subjected to 20 ns MD simulations as described in the simulation section.

#### 3.5.2. Molecular Dynamic (MD) Simulations

The integration of molecular dynamic (MD) simulations into biological systems studies enables exploring the physical motion of atoms and molecules that cannot be easily accessed by any other means. The insight extracted from performing this simulation provides an intricate perspective into the biological systems’ dynamical evolution, such as conformational changes and molecule association [61]. The MD simulations of all systems were performed using the GPU version of the PMEMD engine present in the AMBER 18 package [62].

The partial atomic charge of each compound was calculated with ANTECHAMBER’s General Amber Force Field (GAFF) technique [63]. The Leap module of the AMBER 18 package implicitly solvated each system within an orthorhombic box of TIP3P water molecules within 10 Å of any box edge. The Leap module was used to neutralize each system by incorporating Na^+^ and Cl^−^ counter ions. A 2000-step initial minimization of each system was carried out in the presence of a 500 kcal/mol applied restraint potential, followed by a 1000-step full minimization using the conjugate gradient algorithm without restraints.

During the MD simulation, each system was gradually heated from 0 K to 300 K over 500 ps, ensuring that all systems had the same amount of atoms and volume. The system’s solutes were subjected to a 10 kcal/mol potential harmonic constraint and a 1 ps collision frequency. Following that, each system was heated and equilibrated for 500 ps at a constant temperature of 300 K. To simulate an isobaric-isothermal (NPT) ensemble, the number of atoms and pressure within each system for each production simulation were kept constant, with the system’s pressure maintained at 1 bar using the Berendsen barostat [64].

For 20 ns, each system was MD simulated. The SHAKE method was used to constrain the hydrogen bond atoms in each simulation. Each simulation used a 2 fs step size and integrated an SPFP precision model. An isobaric-isothermal ensemble (NPT) with randomized seeding, constant pressure of 1 bar, a pressure-coupling constant of 2 ps, a temperature of 300 K, and a Langevin thermostat with a collision frequency of 1 ps was used in the simulations.

#### 3.5.3. Post-MD Analysis

After saving the trajectories obtained by MD simulations every 1 ps, the trajectories were analyzed using the AMBER18 suite’s CPPTRAJ [65] module. The Origin data analysis program and Chimera were used to create all graphs and visualizations.

#### 3.5.4. Thermodynamic Calculation

The Poisson–Boltzmann or generalized born and surface area continuum solvation (MM/PBSA and MM/GBSA) approach has been found to be useful in the estimation of ligand-binding affinities [66,67]. The protein–ligand complex molecular simulations used by MM/GBSA and MM/PBSA compute rigorous statistical-mechanical binding free energy within a defined force field.

Binding free energy averaged over 500 snapshots extracted from the entire 50 ns trajectory. The estimation of the change in binding free energy (ΔG) for each molecular species (complex, ligand, and receptor) can be represented as follows [68]:(1)$$\Delta G_{\mathrm{bind}}=G_{\mathrm{complex}}-G_{\mathrm{receptor}}-G_{\mathrm{ligand}}$$
(2)$$\Delta G_{\mathrm{bind}}=E_{\mathrm{gas}}+G_{\mathrm{sol}}-\mathrm{TS}$$
(3)$$E_{\mathrm{gas}}=E_{\mathrm{int}}+E_{\mathrm{vdw}}+E_{\mathrm{ele}}$$
(4)$$G_{\mathrm{sol}}=G_{\mathrm{GB}}+G_{\mathrm{SA}}$$
(5)$$G_{\mathrm{SA}}=\gamma \mathrm{SASA}$$

The terms Egas, Eint, Eele, and Evdw symbolize gas-phase energy, internal energy, Coulomb energy, and van der Waals energy. The Egas was directly assessed from the FF14SB force field terms. Solvation free energy (Gsol) was evaluated from the energy involvement of the polar states (GGB) and non-polar states (G). The non-polar solvation free energy (GSA) was determined from the solvent accessible surface area (SASA) [69,70] using a water probe radius of 1.4 Å. In contrast, solving the GB equation assessed the polar solvation (GGB) contribution. Items S and T symbolize the total entropy of the solute and temperature, respectively.

### 3.6. Statistical Analysis

The presented results were obtained from the mean values of three experimental processes independently performed. CC_50_ and IC_50_ values were determined from the curve of the dose response along with the regression analysis of the triplicates of the values. A one-way ANOVA followed by multiple Tukey’s tests was applied for the comparison of the EOs with *p* < 0.05 significance in the antiviral assay.

## 4. Conclusions

The analysis of the EOs derived from the bark and fruits of *Acacia nilotica* revealed high relative concentrations of terpenoids in both oils. Stachene, caryophyllene oxide, spathulenol, trans-caryophyllene, Z-anethole, and γ-terpinene represented the main constituents. The EO of the bark exhibited moderate anti-HAV and anti-HSV1, while fruit EO showed weak effects against HAV, HSV1, and HSV2. Caryophyllene oxide and spathulenol exhibited the best affinities against the 3C protease and TK proteins. The molecular dynamics simulation proved the significant van der Waals energy of caryophyllene oxide with 3C protease of HAV and thymidine kinase of HSV enzyme. The present findings revealed the effects of the main constituents of *A. nilotica* EO. However, in vivo studies should be evaluated for these EOs and/or their major compounds, either in combination or singular, to determine the actual action mechanisms and safety.

## Figures and Tables

**Figure 1 plants-11-02889-f001:**
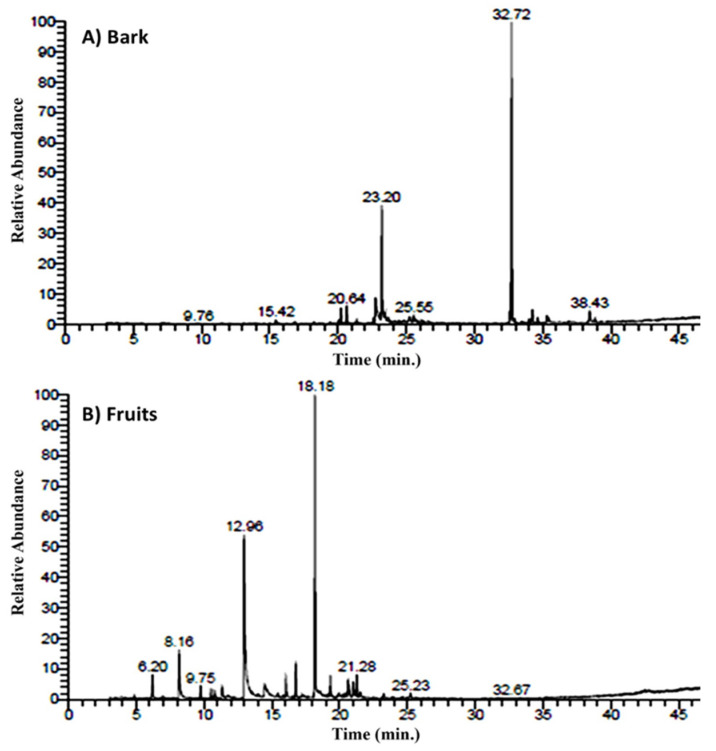
GC-MS ion chromatograms of bark (**A**) and fruits (**B**) of *Acacia nilotica* EOs.

**Figure 2 plants-11-02889-f002:**
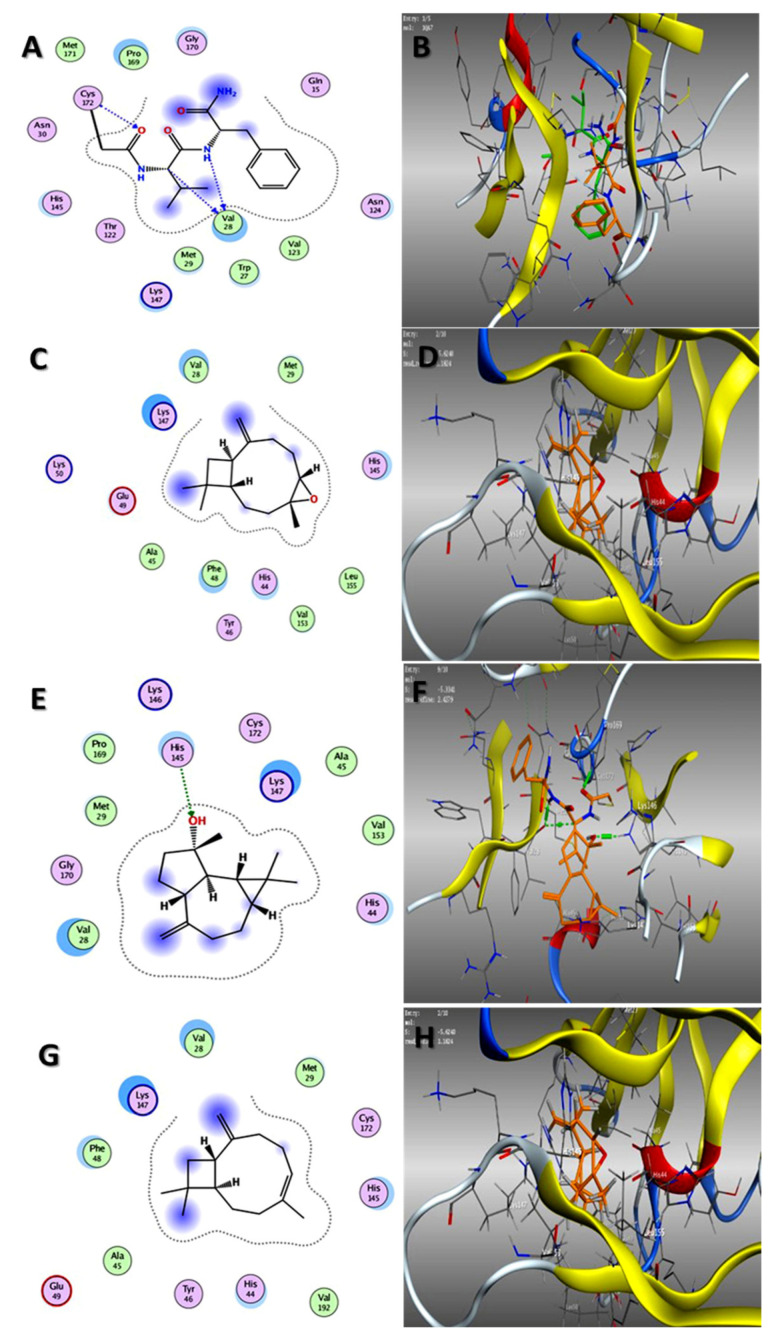
Two-dimensional binding mode of the co-crystallized ligand (**A**) and validation of docking study (**B**), 2D binding mode and 3D binding modes of caryophyllene oxide (**C**,**D**), spathulenol (**E**,**F**), and trans-caryophyllene (**G**,**H**) in the active site of 3C protease of HAV (PDB ID: 1QA7): overlay of the experimental (green) and docked conformation (orange).

**Figure 3 plants-11-02889-f003:**
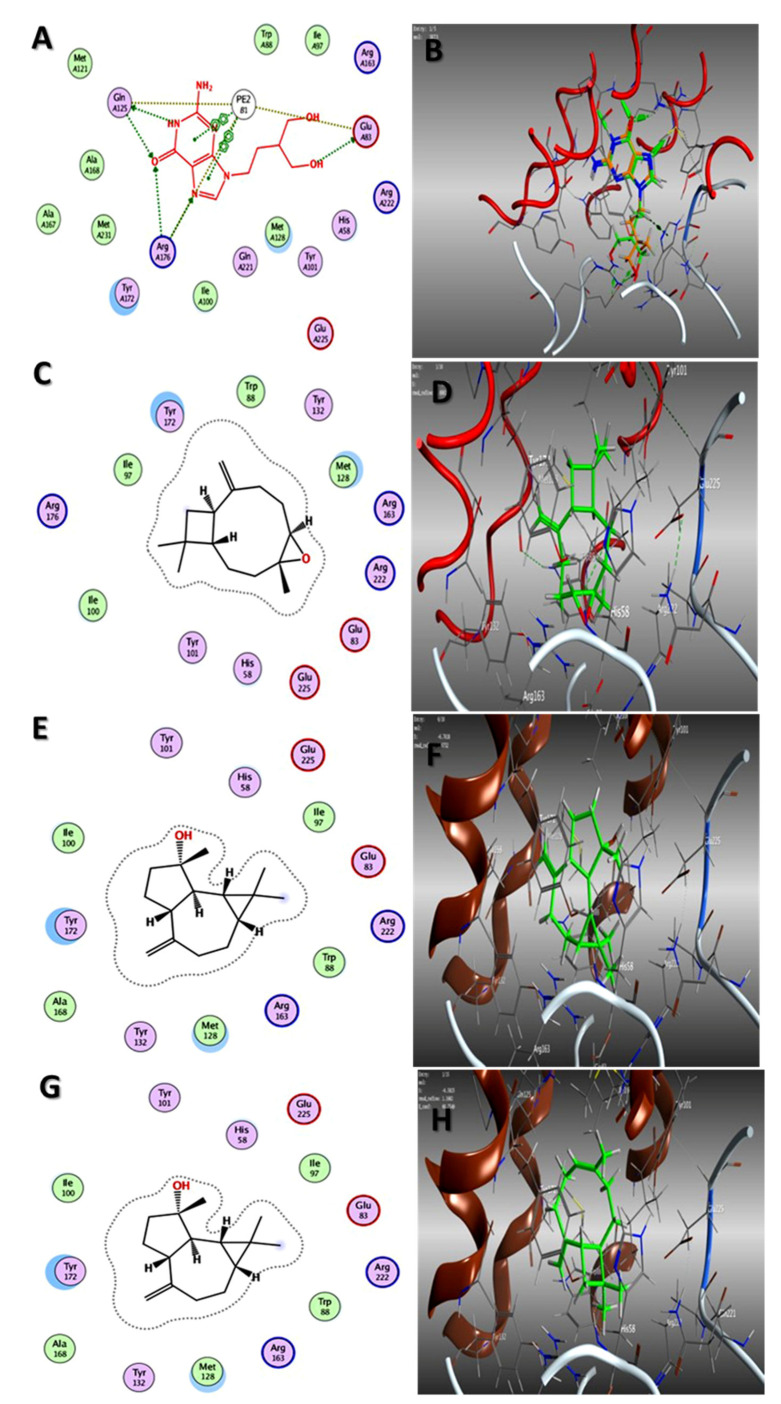
Two-dimensional binding mode of the co-crystallized ligand (**A**) and validation of docking study (**B**), 2D binding mode and 3D binding modes of caryophyllene oxide (**C**,**D**), spathulenol (**E**,**F**), and trans-caryophyllene (**G**,**H**) in the active site of thymidine kinase of HSV (PDB ID: 1KI3): overlay of the experimental (green) and docked conformation (orange).

**Figure 4 plants-11-02889-f004:**
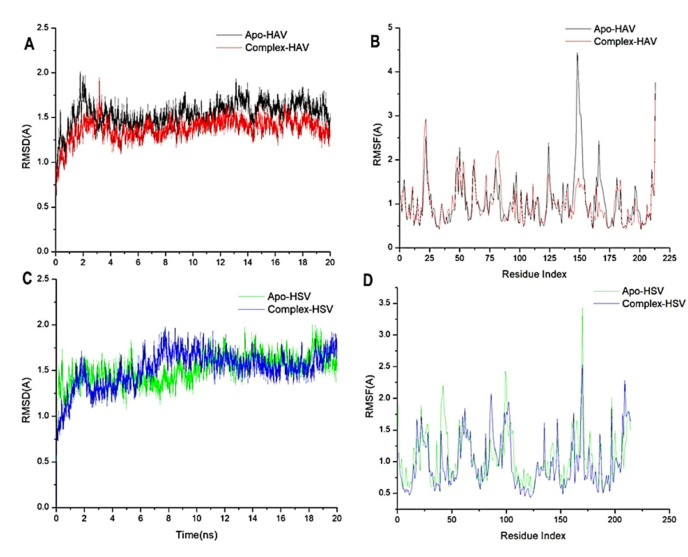
(**A**) RMSD of Cα atoms of the 3C protease of HAV protein backbone atoms. (**B**) RMSF of each residue of the 3C protease of HAV protein backbone Cα atoms (**C**) RMSD of Cα atoms of the thymidine kinase of HSV protein backbone atoms (**D**) RMSF of each residue of the thymidine kinase of HSV protein backbone Cα atoms.

**Table 1 plants-11-02889-t001:** Essential oil compositions of EOs of bark and fruits of *Acacia nilotica*.

No	Rt ^1^	Compound	Type	Relative Concentration (%)		KI	
				Bark	Fruit	Lit. ^2^	Exp. ^3^
1	3.89	α-Pinene	MH	-	0.11 ± 0.01	932	935
2	4.85	β-Pinene	MH	-	0.35 ± 0.02	974	979
3	6.20	1,8-Cineole	OM	-	3.52 ± 0.06	1026	1020
4	6.41	γ-Terpinene	MH	-	7.35 ± 0.08	1054	1059
5	6.63	α-Linalool	OM	-	1.82 ± 0.04	1095	1091
6	6.94	Camphor	OM	-	0.36 ± 0.02	1146	1144
7	9.75	Borneol	OM	-	2.31 ± 0.04	1169	1165
8	10.50	4-Terpineol	OM	-	1.15 ± 0.03	1177	1175
9	10.77	α-Terpineol	OM	-	1.16 ± 0.05	1186	1185
10	11.74	Cumin aldehyde	OM	-	0.49 ± 0.02	1238	1235
11	12.96	*Z*-Anethole	OM	-	22.87 ± 0.23	1249	1245
12	13.99	Bornyl acetate	OM	-	0.23 ± 0.02	1285	1283
13	14.47	2-Caren-10-al	OM	-	3.51 ± 0.05	1289	1287
14	15.45	α-Terpinyl acetate	OM	-	0.47 ± 0.02	1316	1314
15	15.82	Myrtenyl acetate	OM	0.84 ± 0.03	-	1324	1322
16	16.04	α-Elemene	SH	-	4.69 ± 0.06	1335	1339
17	16.13	α-Cubebene	SH	-	0.32 ± 0.01	1351	1346
18	16.54	α-Copaene	SH	0.35 ± 0.02	0.23 ± 0.01	1374	1370
19	16.67	β-Elemene	SH	-	3.72 ± 0.08	1389	1392
20	16.84	Methyl eugenol	OM	-	0.54 ± 0.02	1403	1401
21	18.18	*trans*-Caryophyllene	SH	0.24 ± 0.01	36.95 ± 0.18	1407	1405
22	18.35	Longifolene	SH	-	0.41 ± 0.01	1408	1413
23	18.58	Aromadendrene	SH	-	0.36 ± 0.02	1439	1433
24	19.40	α-Humulene	SH	0.27 ± 0.01	4.05 ± 0.06	1452	1455
25	19.85	γ-Muurolene	SH	-	0.26 ± 0.01	1478	1475
26	19.96	Germacrene-D	SH	0.14 ± 0.01	0.46 ± 0.02	1481	1484
27	20.02	α-Amorphene	SH	0.51 ± 0.02	-	1483	1486
28	20.12	α-Selinene	SH	-	0.54 ± 0.01	1498	1495
29	20.21	α-Muurolene	SH	2.42 ± 0.04	-	1500	1498
30	20.64	Bicyclogermacrene	SH	2.41 ± 0.06	-	1501	1504
31	21.35	δ-Cadinene	SH	0.75 ± 0.02	-	1522	1525
32	22.58	α-Calacorene	SH	1.18 ± 0.05	-	1545	1543
33	22.19	*E*-Nerolidol	OS	0.16 ± 0.01	-	1531	1535
34	22.76	Spathulenol	OS	4.74 ± 0.05	-	1577	1574
35	23.20	Caryophyllene oxide	OS	19.11 ± 0.09	-	1582	1585
36	23.45	Globulol	OS	1.01 ± 0.02	-	1590	1593
37	23.70	Veridiflorol	OS	0.83 ± 0.01	-	1596	1598
38	24.45	Neoclovenoxid	OS	0.55 ± 0.01	-	1608	1605
39	24.68	Isospathulenol	OS	0.14 ± 0.00	-	1630	1632
40	24.85	tau-Cadinol	OS	0.51 ± 0.02	-	1640	1642
42	25.22	Cubenol	OS	0.75 ± 0.03	-	1645	1644
43	25.28	Torreyol	OS	0.68 ± 0.03	-	1646	1648
44	25.55	α-Cadinol	OS	1.35 ± 0.06	-	1654	1657
45	25.68	Khusinol	OS	0.44 ± 0.01	-	1658	1659
46	32.58	Cryptomeridiol	OS	1.27 ± 0.07	-	1813	1816
47	32.72	Stachene	DH	48.34 ± 0.25	-	1931	1934
48	34.26	Trachyloban	DH	2.25 ± 0.07	-	1965	1968
49	34.62	Isokaurene	DH	1.01 ± 0.04	-	1997	1999
50	35.31	Kaur-16-ene	DH	1.94 ± 0.06	-	2017	2015
51	34.02	Phytol	OD	0.76 ± 0.02	-	1942	1940
52	36.42	Sclareol	OD	0.14 ± 0.00	-	2223	2221
53	37.11	4,8,13-Duvatriene-1,3-diol	OD	0.16 ± 0.00	-	2400	2403
54	38.43	*n*-Nonacosane	Others	2.51 ± 0.06	-	2900	2900
56	45.21	*n*-Dotriacontane	Others	0.98 ± 0.03	-	3200	3200
Monoterpene Hydrocarbons (MH)				0	7.81		
Oxygenated Monoterpenes (OM)				0.84	55.51		
Sesquiterpene Hydrocarbons (SH)				8.27	34.91		
Oxygenated Sesquiterpenes (OS)				31.54	3.52		
Diterpene Hydrocarbons (DH)				53.54	0		
Oxygenated Diterpenes (OD)				1.06	0		
Others				3.49	0		
Total				98.74	98.23		

^1^ Retention time (RT), ^2^ Published Kovats indexes (KI_p_), ^3^ Calculated Kovats indexes (KIc).

**Table 2 plants-11-02889-t002:** Effect of EOs of bark and fruits of *Acacia nilotica* on hepatitis A (HAV) and herpes simplex (HSV1 and HSV2).

*Acacia nilotica*	MNTC (µg/mL) ^a^	Antiviral Effect %			Selectivity Index (SI)		
		HAV	HSV1	HSV2	HAV	HSV1	HSV2
Bark EO	500 ± 6.2	47.26 ± 2.05	35.98 ± 1.31	9.07 ± 0.36	2.3	1.6	ND
Fruits EO	1000 ± 11.4	9.42 ± 0.62	14.26 ± 0.54	3.99 ± 0.15	3.8	5.7	1.6
Acyclovir						>387.63	12.24
Amantadine					51.62		

^a^ MNTC: the maximum non-toxic concentration; SI = selectivity index (CC_50_/IC_50_).

**Table 3 plants-11-02889-t003:** Results of docking simulations of the major compounds identified in EOs of both bark and fruits of *Acacia nilotica*.

Plant Part EO	Name of Phytoligands	ΔG * (kcal/mol)	
		HAV 3C Protease	HSV TK
Bark	Spathulenol	−5.23	−6.83
	Caryophyllene oxide	−5.43	−6.96
	Stachene	-	-
Fruits	γ-Terpinene	−4.85	−5.80
	Z-Anethole	−4.66	−5.20
	Trans-caryophyllene	−5.26	−6.79
Co-crystallized inhibitor of HAV 3C protease		−6.61	ND
Co-crystallized inhibitor of HSV TK 1		ND	−7.85

* binding free energy.

**Table 4 plants-11-02889-t004:** The calculated energy binding for caryophyllene oxide against the 3C protease of HAV and thymidine kinase of HSV receptors.

Energy Components (kcal/mol)					
Complex	ΔE_vdW_	ΔE_elec_	ΔG_gas_	ΔG_solv_	ΔG_bind_
Caryophyllene oxide -HAV	−20.44 ± 0.27	−1.52 ± 0.12	−12.96 ± 0.33	3.60 ± 0.11	−19.35 ± 0.24
Caryophyllene oxide -HSV	−34.86 ± 0.081	−2.58 ± 0.07	−37.45 ± 0.11	5.40 ± 0.05	−32.04 ± 0.11

∆EvdW = van der Waals energy; ∆Eele = electrostatic energy; ∆Gsolv = solvation free energy; ∆Gbind = calculated total binding free energy.

## Data Availability

Not applicable.

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
