# Peer review of "Antiviral Potentialities of Chemical Characterized Essential Oils of Acacia nilotica Bark and Fruits against Hepatitis A and Herpes Simplex Viruses: In Vitro, In Silico, and Molecular Dynamics Studies"

_plants, 2022, doi:10.3390/plants11212889_

Round 1

Reviewer 1 Report

The manuscript by Abd El-Nasser G. El Gendy et al. describes the Antiviral potentialities of chemical characterized essential oils of Acacia nilotica bark and fruits against hepatitis A and herpes simplex viruses: in-vitro, in-silico, and molecular dynamics studies.

The research has been carried out properly and the results are of interest

Some minor points should be addressed before acceptance:

The authors have reported the chemical composition of essential oils extracted based on the GC-MS.Despite the large percentage of di-terpenes, the oils show a moderate in vitro Antiviral activities against HAV, HSV1 and HSV2 and a very low SI. May the authors rationalize these results from the numerous data in the literature?

Page 6 Line 157: “disappearancefrom” please should be correct

Page 6 line165-166: the authors reported: “while γ-terpinene (7.35%), and 1,8-cineole (3.52%) are the major monoterpene hydrocarbons” please should be correct because the 1,8 cineole is an oxygenated Monoterpene.

Page 13 line 391: “The formed formazan crystals” please should be correct.

Author Response

Response to Reviewer #1

Dear respected reviewer

               According to your valuable comments, we have revised the manuscript “Antiviral potentialities of chemical characterized essential oils of Acacia nilotica bark and fruits against hepatitis A and herpes simplex viruses: in-vitro, in-silico, and molecular dynamics studies”. In the revised version of the manuscript, we have changed the text providing answers to all the enumerated points. Please, find attached the reviewer’s comments with our point-by-point responses and the revised manuscript. Thank you for your valuable comments that allowed us to improve the manuscript greatly.

Comment: The manuscript by Abd El-Nasser G. El Gendy et al. describes the Antiviral potentialities of chemical characterized essential oils of Acacia nilotica bark and fruits against hepatitis A and herpes simplex viruses: in-vitroin-silico, and molecular dynamics studies. The research has been carried out properly and the results are of interest. Some minor points should be addressed before acceptance:

Response: Thanks for your efforts and time in the revision of our manuscript. We appreciated your comments which will improve the manuscript greatly. The responses to your comments were performed point by point as below.

Comment: The authors have reported the chemical composition of essential oils extracted based on the GC-MS. Despite the large percentage of di-terpenes, the oils show a moderate in vitro Antiviral activities against HAV, HSV1 and HSV2 and a very low SI. May the authors rationalize these results from the numerous data in the literature?

Response: Several data from the literature were inserted and rationalized in the revised version of the manuscript.

Comment: Page 6 Line 157: “disappearance from” please should be correct

Response: “disappearancefrom” was corrected to “disappearance from”

Comment: Page 6 line165-166: the authors reported: “while γ-terpinene (7.35%), and 1,8-cineole (3.52%) are the major monoterpene hydrocarbons” please should be correct because the 1,8 cineole is an oxygenated Monoterpene.

Response: Sorry for this mistake. In the revised version of the manuscript, “1,8 cineole” was revised and corrected within the whole manuscript and all its related concentrations in the Table and the text.

Comment: Page 13 line 391: “The formed formazan crystals” please should be correct.

Response: corrected.

Reviewer 2 Report

The manuscript is well written and can be of interest for researchers involved inthe same field. Some English and mistyping errors throughout the manuscript. Please revise carefully

Author Response

Response to Reviewer #2

Dear respected reviewer

               According to your valuable comments, we have revised the manuscript “Antiviral potentialities of chemical characterized essential oils of Acacia nilotica bark and fruits against hepatitis A and herpes simplex viruses: in-vitro, in-silico, and molecular dynamics studies”. In the revised version of the manuscript, we have changed the text providing answers to all the enumerated points. Please, find attached the reviewer’s comments with our point-by-point responses and the revised manuscript. Thank you for your valuable comments that allowed us to improve the manuscript greatly.

Comment: The manuscript is well written and can be of interest for researchers involved in the same field. Some English and mistyping errors throughout the manuscript. Please revise carefully

Response: We appreciated your time and efforts in the revision of our manuscript. The overall manuscript was carefully revised and corrected properly.

Reviewer 3 Report

Dear authors,

Please pay attention to the following items:

- The first time citing a species, use the full botanical synonym, including the authority (e.g., title and lines 100-101). For other species than from the Acacia genus, include, also, the family

-results and discussion: elaborate better on the literature finds on the antiviral activity of the major compounds in each EO

- antiviral assays: as the volatile oil was separated from hydrolate using n-hexane, please inform if a solvent control was used in all the in vitro assay

Author Response

Dear respected reviewer

               According to your valuable comments, we have revised the manuscript “Antiviral potentialities of chemical characterized essential oils of Acacia nilotica bark and fruits against hepatitis A and herpes simplex viruses: in-vitro, in-silico, and molecular dynamics studies”. In the revised version of the manuscript, we have changed the text providing answers to all the enumerated points. Please, find attached the reviewer’s comments with our point-by-point responses and the revised manuscript. Thank you for your valuable comments that allowed us to improve the manuscript greatly.

Comment: Dear authors, Please pay attention to the following items:

The first time citing a species, use the full botanical synonym, including the authority (e.g., title and lines 100-101). For other species than from the Acacia genus, include, also, the family

Response:

  • We appreciated your time and efforts in the revision of our manuscript. We also thank you for your valuable comments and corrections that will improve our manuscript greatly.
  • All the plants` names were revised and the authority was provided for the first mention either in the abstract or the text. Also, the family of the species was provided. However, for the title, we preferred not to mention the authority as it will become too long as “Acacia nilotica (L.) P.J.H.Hurter & Mabb.”. We hope this meets your acceptance.

Comment: -results and discussion: elaborate better on the literature finds on the antiviral activity of the major compounds in each EO

Response: The antiviral of the major compounds was elaborated from the literature in the sentence. Please, check the revised version of the manuscript.

Comment: - antiviral assays: as the volatile oil was separated from hydrolate using n-hexane, please inform if a solvent control was used in all the in vitro assay

Response: Thanks for your clarification. We used DMOS as a solvent, and this was used as a control, where it showed no activity. This point was clarified in the experimental section of the revised manuscript. Please, check the revised manuscript.